# Interaction of Shock Waves with Discrete Gas Inhomogeneities: A Smoothed Particle Hydrodynamics Approach

**Andrea Albano** *,† and **Alessio Alexiadis** †

School of Chemical Engineering, University of Birmingham, Birmingham B15 2TT, UK; Alexiadis@bham.ac.uk

\* Correspondence: AXA1220@student.bham.ac.uk

† These authors contributed equally to this work.

**Abstract:** In this study, we propose a smoothed particle hydrodynamics model for simulating a shock wave interacting with cylindrical gas inhomogeneities inside a shock tube. When the gas inhomogeneity interacts with the shock wave, it assumes different shapes depending on the difference in densities between the gas inhomogeneity and the external gas. The model uses a piecewise smoothing length approach and is validated by comparing the results obtained with experimental and CFD data available in the literature. In all the cases considered, the evolution of the inhomogeneity is similar to the experimental shadowgraphs and is at least as accurate as the CFD results in terms of timescale and shape of the gas inhomogeneity.

**Keywords:** particle method; smoothed particle hydrodynamics; modelling; simulations; shock wave

---

## 1. Introduction

In the last 30 years, the study of a planar shock wave interacting with an isolated, gas inhomogeneity has been investigated both experimentally (e.g., [1–5]) and numerically (e.g., [4,6–8]). Nowadays, this system has acquired importance for computational models up to the point of becoming a benchmark for validating shock-induced flows [9].

A gas inhomogeneity is created in a tube (known as shock tube) filled with gas by slowly introducing a different type of gas. The shock tube is generally a tube with either a rectangular or circular cross section; the shock wave can be generated either from an explosion (blast-driven) or due to high-pressure differences between two gasses separated by a diaphragm (compressed gas-driven). When the diaphragm breaks out, a shock wave is generated and propagates through the gas at lower pressure. In Figure 1, the lower pressure gas is called driven gas and the high-pressure gas that generates the shock wave driver gas. Analogously, the section of the tube where the driver gas is confined is called driver section while the section of the driven gas is called driven section. As a result of the shock wave, a net flow, in the direction of the shock wave but with lower speed, is generated. In rectangular shock tubes, mixing between the two gasses is initially avoided by injecting the inhomogeneity in a nitrocellulose membrane (cylindrical inhomogeneity). In this way the inhomogeneity remains "cylindrical" during its evolution (Figure 2). When the shock wave reaches the inhomogeneity, the inhomogeneity deforms in a way that depends on the density difference between the inhomogeneity and the driven gas. The different shapes that the inhomogeneity assumes during the passage of the shock wave are typically used for validation of numerical codes (e.g., [6,10]).

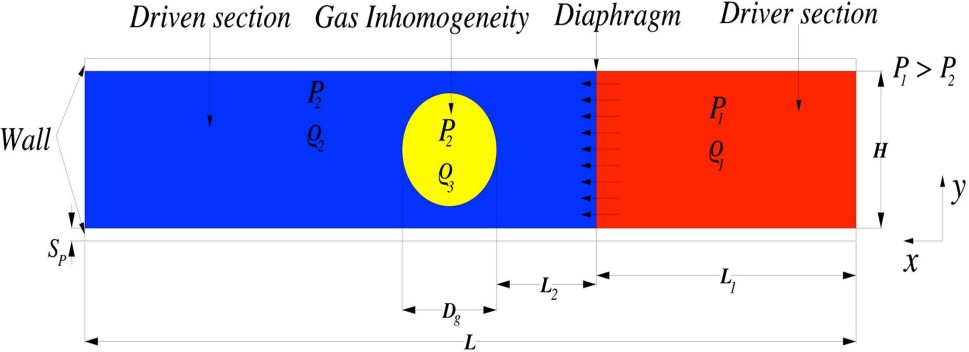

**Figure 1.** Geometry of the simulation box.

In this work, we use smoothed particle hydrodynamics (SPH) to simulate the shock wave and the inhomogeneity interaction. The different shapes of the inhomogeneity calculated during the simulation are compared with experimental data available in the literature for assessing the precision of the model.

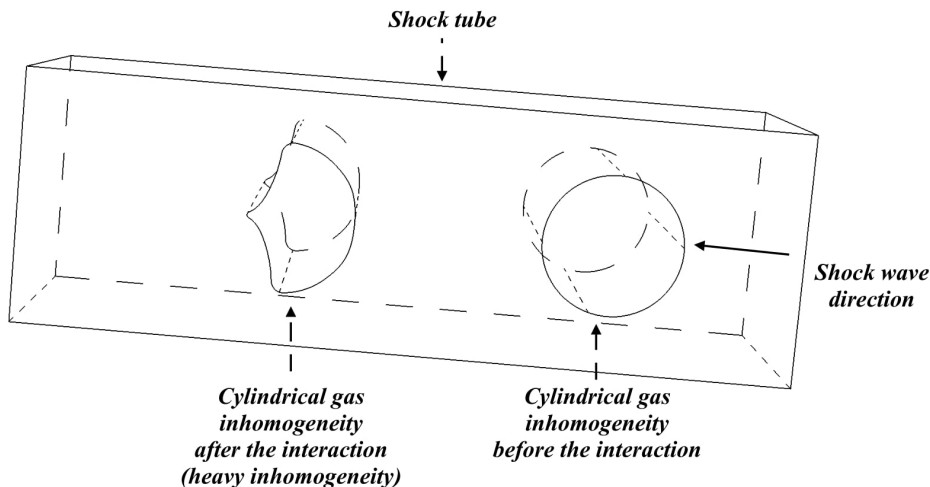

**Figure 2.** Geometry of the shock tube with the cylindrical inhomogeneity.

## 2. Smoothed Particle Hydrodynamics

Smoothed particle hydrodynamics is a meshfree computational method initially developed by Gingold and Monaghan [11] and Lucy [12] for solving astrophysical problems. Later it was used to solve fluidynamics problems to overcome some of the limitations of the grid-based method in the case, for instance, of explosions and high velocity impact phenomena ([13,14]). The method was also widely validated against shock waves in particular for the well known Riemann problem ([15–17]). The SPH approximation is based on the so-called integral representation of a function. Given the function $f(\mathbf{r})$, defined in a volume $V$, function of the three-dimensional position $\mathbf{r}$, is defined by the identity

$$f(\mathbf{r}) = \iiint f(\mathbf{r}')\delta(\mathbf{r} - \mathbf{r}')d\mathbf{r}', \tag{1}$$

where $\delta(\mathbf{r} - \mathbf{r}')$ is the Dirac delta function defined as

$$\delta(\mathbf{r} - \mathbf{r}') = \begin{cases} +\infty & \mathbf{r} = \mathbf{r}' \\ 0 & \mathbf{r} \neq \mathbf{r}'. \end{cases} \tag{2}$$

It is possible to approximate the integral representation by replacing the Dirac delta function with a bell-shaped function called smoothing function or kernel, $W$, which depends on the position $\mathbf{r}$ and on the so-called smoothing length, $h$. When $h$ approaches zero the kernel function $W$ has the property

$$\lim_{h \to 0} = W(\mathbf{r} - \mathbf{r}', h) = \delta(\mathbf{r} - \mathbf{r}'), \tag{3}$$

which approximates the integral representation, Equation (1), to the so-called kernel approximation:

$$f(\mathbf{r}) \approx \iiint f(\mathbf{r}')W(\mathbf{r} - \mathbf{r}', h)d\mathbf{r}'. \tag{4}$$

In this work, the kernel used is Lucy kernel function

$$W(R, h) = \begin{cases} \frac{1}{s} \left[1 + 3\frac{R}{h}\right] \left[1 - \frac{R}{h}\right]^3 & R \leq 1 \\ 0 & R > 1, \end{cases} \tag{5}$$

where $R = |\mathbf{r} - \mathbf{r}'|$ and $s$ is a parameter used to normalise the kernel function, which, for one, two and three dimensional space is, respectively, $\frac{4h}{5}$, $\frac{\pi h^2}{5}$ and $\frac{16\pi h^3}{105}$. The last step is to approximate the infinitesimal volume $d\mathbf{r}'$ to a finite volume $dr$ composed by computational particles with their own mass $m = \rho dr$. With this approximation it is possible to discretise Equation (4)

$$f(\mathbf{r}) \approx \sum \frac{m_i}{\rho_i} f(\mathbf{r}_i)W(\mathbf{r} - \mathbf{r_i}, h), \tag{6}$$

where $m_i$, $\rho_i$ and $\mathbf{r}_i$ are mass, density and position of the $i$th particle. Only particles for which $|\mathbf{r} - \mathbf{r_i}| < h$ are taken into account in the summation. With Equation (6) it is possible to discretise any set of equations such as the energy balance or the Navier–Stokes equation on an arbitrarily set of computational particles. Within the SPH framework, for instance, the momentum–conservation equation can be rewritten as

$$m_i \frac{d\mathbf{v}_i}{dt} = \sum_j m_i m_j \left(\frac{P_i}{\rho_i} + \frac{P_i}{\rho_i} + \Pi_{ij}\right) \nabla_j W_{ij}, \tag{7}$$

where $W_{ij} = W(r_j - r_i, h)$ and $\nabla_j W_{ij}$ is the kernel gradient in the $r_j$ direction. $P$ is the pressure while $\Pi_{ij}$ is the so-called artificial viscosity introduced by Monaghan [18] for simulating shock waves

$$\Pi_{ij} = -\alpha h \frac{c_i + c_j}{\rho_i + \rho_j} \frac{\mathbf{v}_{ij} \cdot \mathbf{r}_{ij}}{r_{ij}^2 + \epsilon h^2}, \tag{8}$$

where $c_i$ and $c_j$ are speed of sound of particles $i$ and $j$ and $\alpha$ is dimensionless parameter that controls the strength of the viscous dissipation and $\epsilon \approx 0.01$ is used to avoid singularities when particles are close to each other. The parameter $\alpha$ can be linked to the kinematic viscosity by means of

$$\nu = \frac{\alpha h c}{8}. \tag{9}$$

During the simulation, Equation (7) updates, at every time step, the velocities of the Lagrangian particles; the density is updated by the continuity equation in discrete form

$$\frac{d\rho_i}{dt} = \sum_j m_j \mathbf{v}_{ij} \nabla_j W_{ij}, \tag{10}$$

where $\mathbf{v}_{ij} = \mathbf{v}_i - \mathbf{v}_j$. Equation (10) requires a closure term relating $\rho$ and $P$. In this work, the ideal gas equation of state is used

$$P(\rho, e) = (\gamma - 1)\rho e, \tag{11}$$

where $\gamma = \frac{C_p}{C_v}$ is the capacity heat ratio and $e$ is the specific internal energy.

## 3. Dimensional Groups

The interaction between the shock wave and the gas inhomogeneity depends on the physical properties of the driven and inhomogeneity gasses and on the shock wave speed, which can be represented as dimensionless groups. In this section, we define the dimensionless groups used in this study.

### 3.1. Atwood Number

The Atwood number is defined as:

$$A = \frac{\rho_3 - \rho_2}{\rho_3 + \rho_2}, \tag{12}$$

where $\rho_2$ and $\rho_3$ are the density of the gas inhomogeneity and air respectively (Figure 1). The Atwood number expresses the interaction between the gas inhomogeneity and the planar incident shock wave. When $A < 0$, we have a "light inhomogeneity", where the inhomogeneity density is lower than that of the driven gas. When $A > 0$, on the contrary, we have a "heavy inhomogeneity", where the inhomogeneity density is higher than that of the driven gas.

### 3.2. Pressure Ratio

The pressure ratio is defined as

$$P_r = \frac{P_1}{P_2} \tag{13}$$

and represents the pressure ratio between the driver gas, $P_1$, and the driven gas, $P_2$.

*3.3. Mach Number*

An important factor that influences the dynamics in the system is the speed with which the shock wave propagates in the driven gas. The shock wave propagates with a speed greater than the sound speed in the fluid expressed as dimensionless Mach number

$$\mathrm{Ma} = \frac{U_s}{c_2},$$

(14)

where $U_s$ is the speed of the shock wave and $c_2$ is the speed of sound of the driven gas.

*3.4. Dimensionless Time*

The last dimensionless number used in this work is the dimensionless time $\tau$, defined as

$$\tau = \frac{t}{\tau_0},$$

(15)

where $t$ is the time of the simulation and $\tau_0$ is the time required for the shock wave to pass through the gas inhomogeneity.

**4. Shape Analysis**

In this section, we look at how, according to the literature, the shape of the inhomogeneity changes with the Atwood number.

*4.1. Standard Shapes for Light Inhomogeneity ($A < 0$)*

Due to the higher sound speed in the gas inhomogeneity, the shock wave finds less resistance and thus moves faster than in the driven gas. At first, the gas inhomogeneity flattens in the direction of the shock wave ($x$-direction according to Figure 1) and expands in the $y$-direction gaining a Semi-prolate shape (Figure 3a). Later a re-entrant jet forms at the centre of the inhomogeneity (crescent moon shape, Figure 3b) and, as it grows, the inhomogeneity changes shapes first to semi claw shape (Figure 3c) and finally to claw shape (Figure 3d).

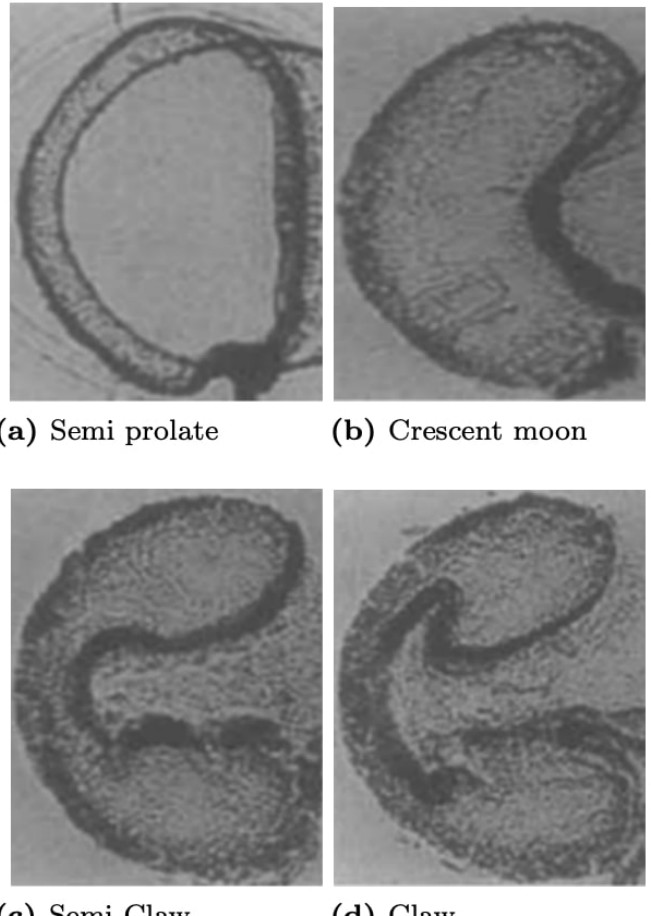

(a) Semi prolate      (b) Crescent moon

(c) Semi Claw      (d) Claw

**Figure 3.** Standard gas inhomogeneity shapes definition using the experimental shadowgraphs [1] for Atwood number: $A = -0.79$. Times: (**a**) 102 μs (**b**) 245 μs (**c**) 427 μs (**d**) 674 μs.

*4.2. Standard Shapes for Heavy Inhomogeneity ($A > 0$)*

When $A > 0$, the speed of sound in the gas inhomogeneity is slower than in the driven gas leading to completely different shapes. Initially the compression effect is predominant and the gas inhomogeneity tends to flatten (flatfish shape, Figure 4a) followed by a crescent shape (jellyfish head shape, Figure 4b). Later, the passage of the shock wave front causes the formations of filaments at the top and bottom of the inhomogeneity (Jellyfish shape, Figure 4c).

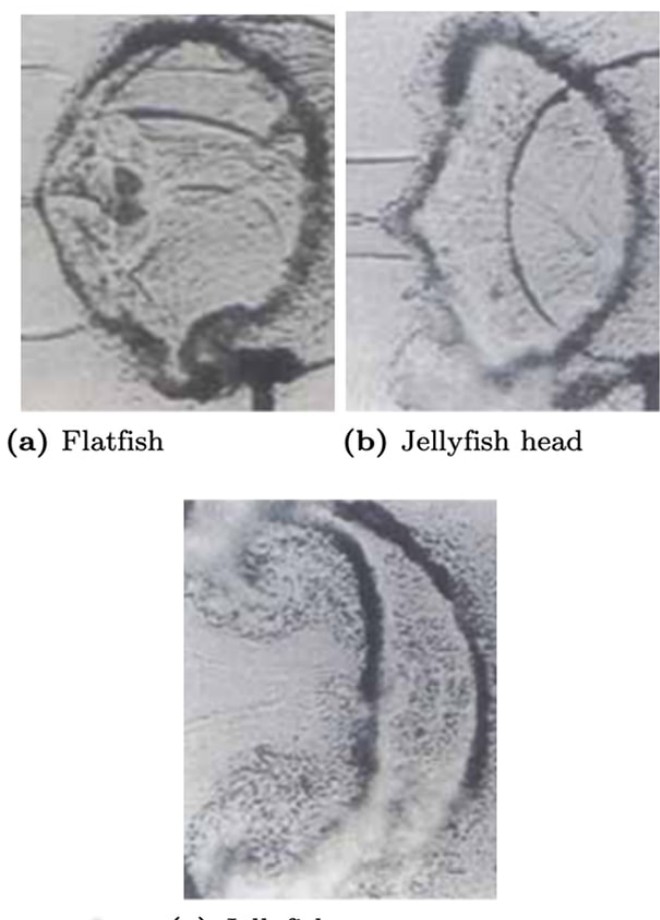

**(a)** Flatfish **(b)** Jellyfish head

**(c)** Jellyfish

**Figure 4.** Standard gas inhomogeneity shapes definition using the experimental shadowgraphs [1] for Atwood number: $A = 0.51$. Times: **(a)** 247 µs **(b)** 417 µs **(c)** 1020 µs.

## 5. Model

### 5.1. Geometry

A simplified 2D replica of a shock tube was developed and is shown in Figure 1. The dimensions of the shock tube and of the gas inhomogeneity were chosen to match the experimental set-up by Haas & Sturtevant [1], who used a rectangular cross section shock tube and a cylindrical gas inhomogeneity, with (referring to Figure 1) $D_g = 0.50$ m, $H = 0.89$ m, $L_1 = 0.50$ m, $L_2 = 0.0225$ m, $L = 1.00$ m and $s_p = 0.00445$ m. In the simulation four types of SPH particles are accounted for: type 1 particles are in the driver gas group, type 2 in the driven gas group, type 3 in the gas inhomogeneity group and type 4 in the wall group. The wall interacts with the fluid with a repulsive force by using the Lennard-Jones potential to avoid compenetration between the fluid and the walls. Along the $x$-axis the boundaries are set as shrink-wrapping. Shrink-wrapping (SW) boundaries are non-periodic boundaries, where the edge of the simulation box moves with the expanding atoms to make sure that all the particles remain within the computational domain [17]. Because in our simulations we only model a section of the tube, SW is used to make sure the shock wave is not reflected when it reaches the boundary. Preliminary simulations were carried out with different resolutions (e.g., total number of particles $N$ = 175,050, 280,450, 565,577, 750,100). The value of $N$ = 375,050 was chosen as the best compromise between accuracy and computational speed.

*5.2. Shock Wave Generation*

In the experiments ([1,3,6,19]), the pressure ratio $P_r$ used to generate the shock wave is not specified. For this reason, we initially run several simulations with the goal to determine which $P_r$ brings to a shock wave with Ma $= 1.22$, as in Haas [1] and Quirk [6].

Firstly, we used a standard single smoothing length approach, where $h$ is greater than the initial particles spacing dL (various solutions with $h$ between 1.05 dL and 1.2 dL were investigated). In this way, however, the speed of the shock wave is always higher than the experimental one. This seems to be a recurring issue in SPH simulations and often the actual speed of the shock wave is not well addressed in the SPH literature. To address a similar issue, SPH simulations of explosions, where the Mach number can reach values of 8.5, often adopt a variable smoothing length changing with the density of the particles in the domain (e.g., [20,21]). In this study, however, considering that the Mach number is lower, a simpler approach based on a piecewise constant smoothing length is adopted here. To achieve correct Mach numbers in our simulations, the first smoothing length, $h_1 >$ dL, is used to simulate the interaction of the particles of the driver section and at the interface between particles of type 1 and particles of type 2. The second smoothing length, $h_2 <$ dL, is used for the particles in the driven section. From a physical point of view, $h_2 <$ dL reflects the fact that the speed of sound is the speed at which a perturbation can move in a fluid. Therefore, the computational particles in the driven section should not "feel" the presence of the shock wave before the passage of the front. From the theoretical point of view, the relation between $h_2$ and *Ma* deserves more investigation. The issue, however, is beyond the scope of the present study; here we simply identified the value of $h_2$ that brings to the correct Mach number; further analysis is left for future work. The correct speed of the shock wave (i.e., Ma $= 1.22$) was achieved with $P_r = 20$, $h_1 = 1.15$ dL and $h_2 = 0.5$ dL. The dissipation factor was chosen to be $\alpha = 0.1$ as in Morris & Monaghan [22].

## 6. Result and Discussion

In the literature, comparison between numerical and experimental data is usually done by comparing the shapes of the inohomogenity at different Atwood numbers. This study follows the same approach and the model is assessed by comparing our simulations with the experimental shadowgraphs reported by Haas [1]. Additionally, we also validate our results with Quirk's CFD simulations [6].

*6.1. Standard Shapes Comparison*

### 6.1.1. Light Inhomogeneity Cases ($A = -0.79$)

Simulation parameters used are shown in Table 1 for the pair Air/Helium ($A = -0.79$) as in [1,6]. For the time step we use the CFL criterion ([23,24]) and $Dt = 10^{-9}s$.

**Table 1.** Computational set-up for the Air-Helium system: $N_{p,1}$, $N_{p,2}$, $N_{p,3}$, $N_{p,4}$ are the number of particles of type 1, 2, 3 and 4. $\rho_1$, $\rho_2$ and $\rho_3$ are the density of particles of type 1, 2, and 3 expressed as Kg m$^{-3}$. $P_r$ is the pressure ratio. $h_1$ is the smoothing length of particles type 1. $h_2$ is the smoothing length of particles type 2 and 3. $\tau_s$ is the dimensionless time step of the simulation. $\alpha$ is the dimensionless factor controlling the dissipation strength. *Ma* is the shock wave Mach number in the simulation.

| $N_{p,1}$ | $N_{p,2}$ | $N_{p,3}$ | $N_{p,4}$ | $\rho_1$ | $\rho_2$ | $\rho_3$ | $P_r$ | $h_1$ | $h_2$ | $\tau_s$ | $\alpha$ | **Ma** |
|---|---|---|---|---|---|---|---|---|---|---|---|---|
| 125,050 | 244,464 | 5536 | 42,121 | 5 | 1.16 | 0.18 | 20 | 1.15 dL | 0.5 dL | $1.39 \cdot 10^{-11}$ | 0.1 | 1.22 |

Comparison between Figures 3 and 5 shows that SPH is in good agreement with the experimental shadowgraphs. With respect to the CFD simulations, the SPH results are slightly less reliable at initial

times (e.g., the first two shapes in Figure 5) but more reliable at later times. The relatively small differences in timescale are discussed in the next section.

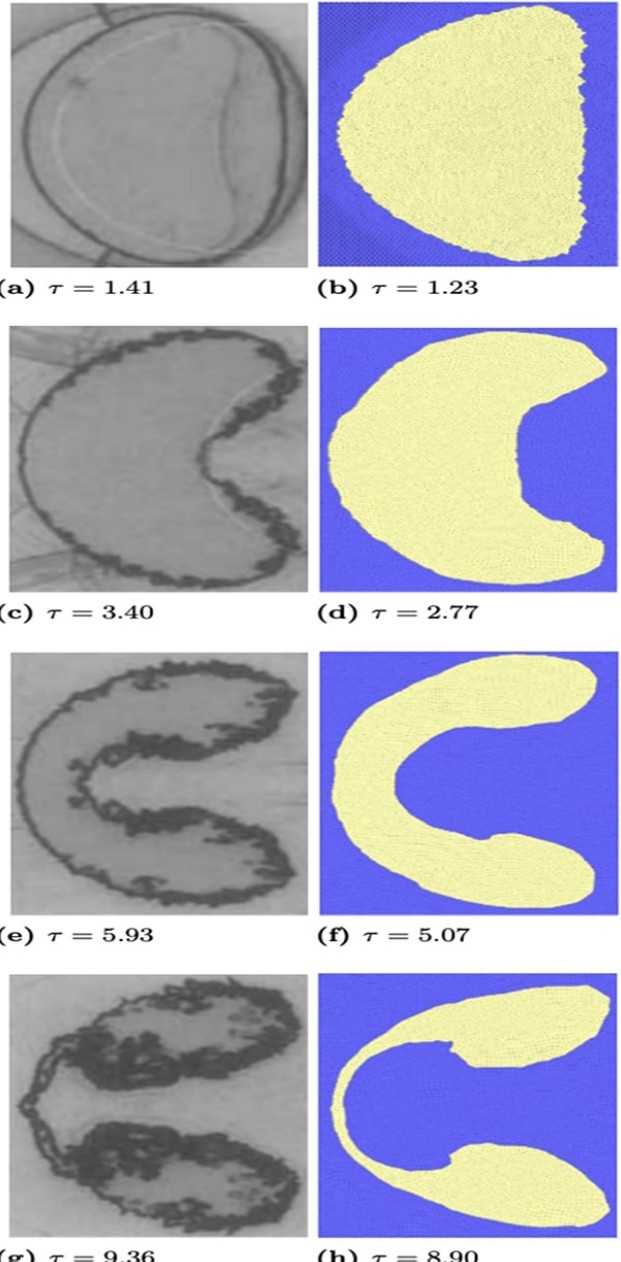

(a) $\tau = 1.41$  (b) $\tau = 1.23$

(c) $\tau = 3.40$  (d) $\tau = 2.77$

(e) $\tau = 5.93$  (f) $\tau = 5.07$

(g) $\tau = 9.36$  (h) $\tau = 8.90$

**Figure 5.** Shape comparison between results [6] (left column) and smoothed particle hydrodynamics (SPH) results (right column) for $A = -0.79$. See Figure 3 for the equivalent experimental data [1].

6.1.2. Heavy Inhomogeneity Cases ($A = 0.51$)

Simulations parameters are shown in Table 2 for the pair Air/Dichlorodifluoromethane (R12) ($A = 0.51$) as in [1,6]. The timestep is $Dt = 10^{-9}$ s as before.

**Table 2.** Computational set-up for the Air-R12 system: $N_{p,1}$, $N_{p,2}$, $N_{p,3}$, $N_{p,4}$ are the number of particles of type 1, 2, 3 and 4. $\rho_1$, $\rho_2$ and $\rho_3$ are the density of particles of type 1, 2, and 3 expressed as Kg m$^{-3}$. $P_r$ is the pressure ratio. $h_1$ is the smoothing length of particles type 1. $h_2$ is the smoothing length of particles type 2 and 3. $\tau_s$ is the dimensionless time step of the simulation. $\alpha$ is the dimensionless factor controlling the dissipation strength. $Ma$ is the shock wave Mach number in the simulation.

| $N_{p,1}$ | $N_{p,2}$ | $N_{p,3}$ | $N_{p,4}$ | $\rho_1$ | $\rho_2$ | $\rho_3$ | $P_r$ | $h_1$ | $h_2$ | $\tau_s$ | $\alpha$ | $Ma$ |
|---|---|---|---|---|---|---|---|---|---|---|---|---|
| 125,050 | 244,464 | 5536 | 42,121 | 5 | 1.16 | 3.65 | 20 | 1.15 dL | 0.5 dL | $1.39 \cdot 10^{-11}$ | 0.1 | 1.22 |

In addition, in this case, the SPH results (Figure 6) show good agreement with the available experimental data (Figure 4). With respect to the CFD simulations, the SPH results look more reliable especially at longer times (e.g., the jellyfish head shape in the last figure of Figure 6). Again, the small timescale differences are discussed in the next section.

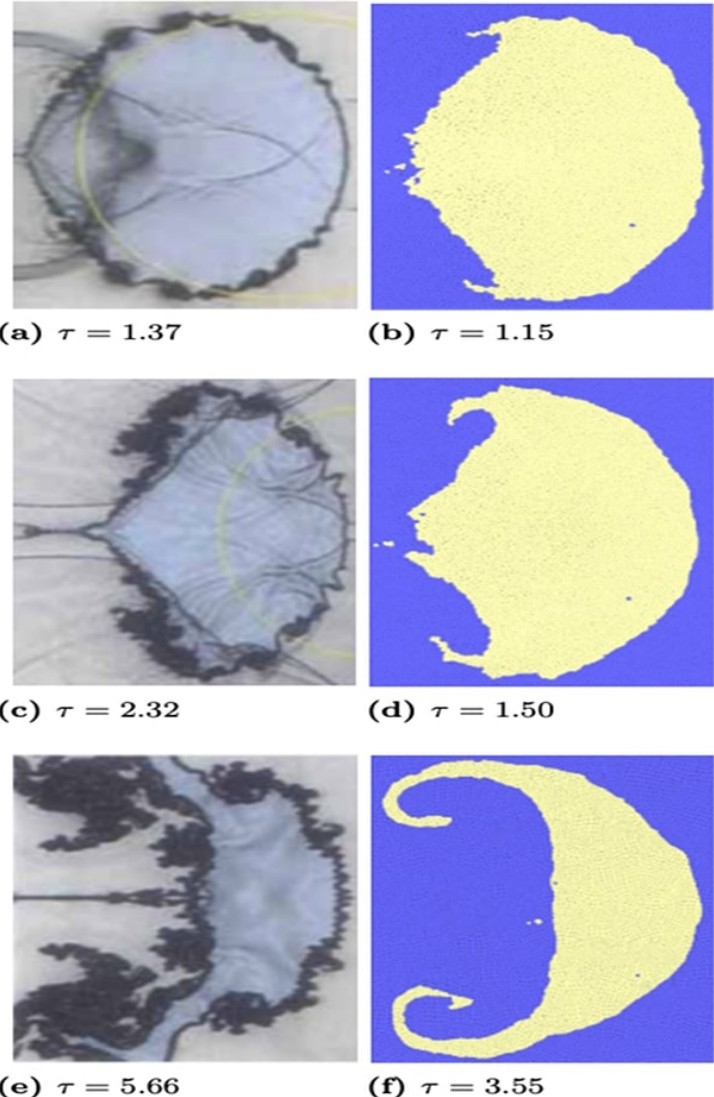

(a) $\tau = 1.37$  (b) $\tau = 1.15$

(c) $\tau = 2.32$  (d) $\tau = 1.50$

(e) $\tau = 5.66$  (f) $\tau = 3.55$

**Figure 6.** Shape comparison between CFD results [6] (**left** column) and SPH results (**right** column) for $A = 0.51$. See Figure 4 for the equivalent experimental data [1].

## 6.2. Timescale Comparison

Our results and those of Quirk [6] show approximately the same shapes at slightly different computational times. In this section we will compare the timescale from different experiments to verify the validity of our timescale. In fact, the identification of the different shapes is usually performed visually and, therefore, a certain inaccuracy is expected. The data presented in Figure 7 are from Haas & Sturtevant [1], Levy [19] and Layes [7,25] and refer to conditions analogous to our SPH model. Results are only presented for the air-helium system. Since the R12 is toxic, the air-R12 system is less investigated and there are not enough data in literature for an exhaustive comparison.

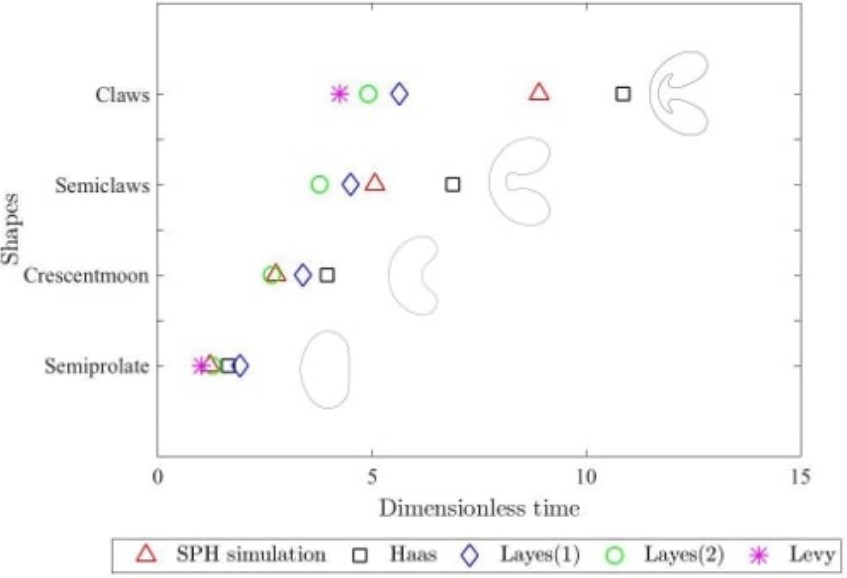

**Figure 7.** Standard shapes timescale comparison.

Figure 7 shows the deviation of experimental data and that our results lie within the experimental uncertainty.

## 6.3. Time Depending Artificial Viscosity

Normally, in SPH simulation, the dissipation constant $\alpha$ is maintained constant during the simulation. However, as a means of improving results in the case of shock waves, Morris & Monaghan [22] introduced a time-varying coefficient, $\alpha(t)$,

$$\alpha(t) = \alpha^* + \alpha_0 exp\left(-\frac{t}{\tau_e}\right), \tag{16}$$

where $\alpha^*$ is the minimum dissipation factor, $\alpha_0$ is the initial dissipation factor and $\tau_e$ is the e-folding time. In this section, we test the variable dissipation of Equation (16) with $\alpha_0 = 0.1$, $\alpha^* = 10^{-6}$ and, following the procedure of Morris & Monaghan [22], $\tau_e = 1.05 \cdot 10^{-5}$, to assess if it can further improve the results. Figures 8 and 9 show the comparison between the results calculated with $\alpha = 0.1$ and $\alpha(t)$. The results at lower times are almost identical to those in Figures 5 and 6 and are not reported. At higher computational times there is a small improvement, especially for the claw shape. However, in general the results with constant $\alpha$ seem reasonably accurate.

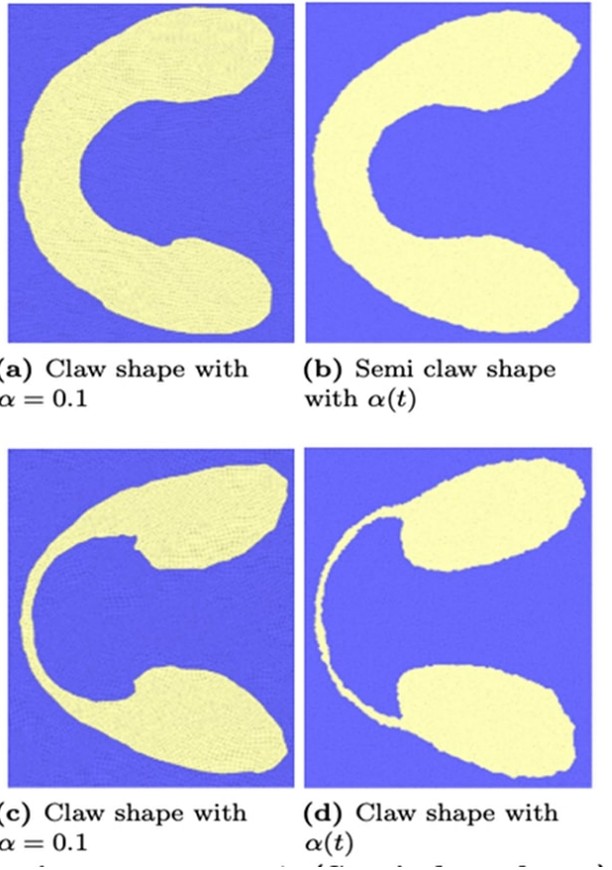

(a) Claw shape with $\alpha = 0.1$

(b) Semi claw shape with $\alpha(t)$

(c) Claw shape with $\alpha = 0.1$

(d) Claw shape with $\alpha(t)$

**Figure 8.** Shapes comparison at $\tau = 5.07$ (Semi claw shape) and $\tau = 8.90$ (Claw shape) between constant viscosity (**left**) and time-varying viscosity (**right**).

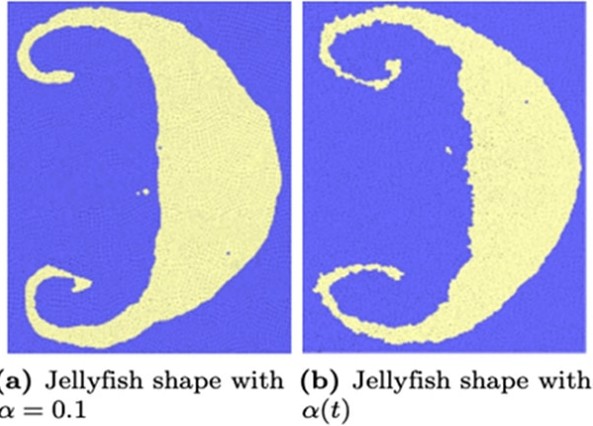

(a) Jellyfish shape with $\alpha = 0.1$

(b) Jellyfish shape with $\alpha(t)$

**Figure 9.** Shape comparison at $\tau = 3.55$ (Jellyfish shape) between constant viscosity (**left**) and time-varying viscosity (**right**).

*6.4. Pressure Field*

This section reports the pressure field calculated with the method proposed in this study for both the light and the heavy inhomogeneity. Analysis of the pressure field, in fact, allows to understand and explain why the inhomogeneity assumes specific shapes during its evolution. In the case of light inhomogeneity

(Figure 10), initially the incident shock wave impacts and reflects on the inhomogeneity (Figure 10a) generating a lower pressure reflected-wave behind the inhomogeneity. At a later stage, a high-pressure region behind the inhomogeneity (Figure 10b) is observed and, consequently, the lighter gas moves away from the high-pressure area giving a crescent moon shape to the inhomogeneity. In the case of heavy inhomogeneity (Figure 11), a reflected-wave (Figure 11a) and a high pressure region (Figure 11b) also forms, but, this time, the reflected wave has a higher pressure than the surroundings and the high-pressure area forms in front of the inhomogeneity. Moreover, the high pressure area is partly outside and partly inside the gas inhomogeneity. As a result of this, the tip of the inhomogeneity stretches, forming the central wedge typical of the jellyfish head shape.

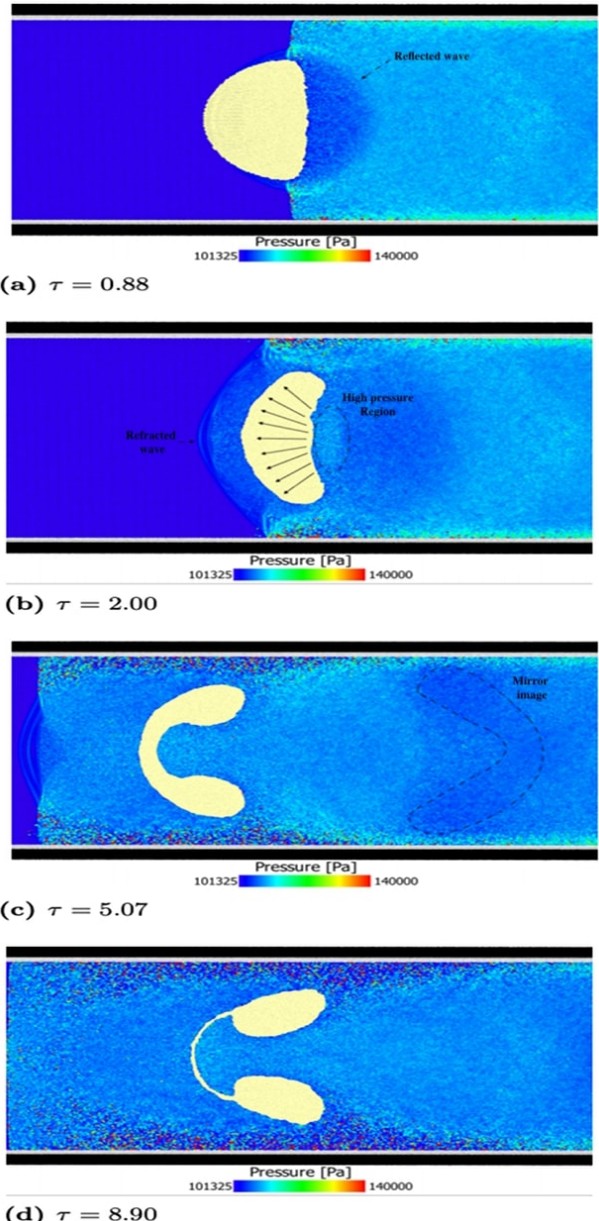

**Figure 10.** Pressure field in the driven gas for the light inhomogeneity ($A = -0.79$) case at different dimensionless times.

The included video files Video1.avi and Video2.avi, see Supplementary Materials, show the evolution of the pressure field at the same conditions, respectively, of Figures 10 and 11.

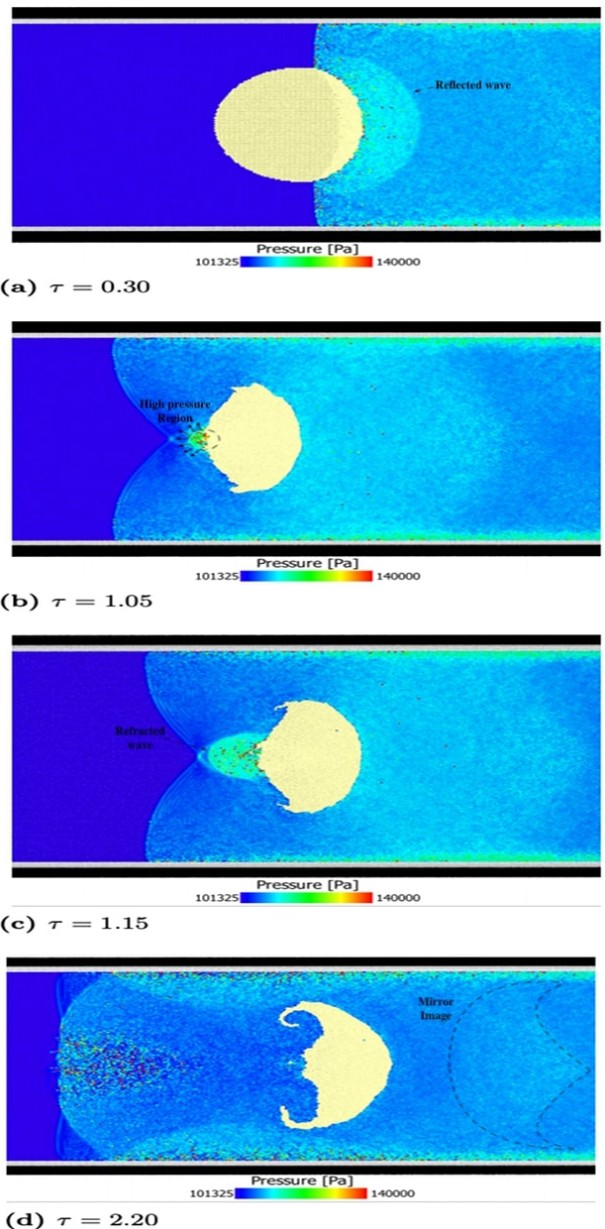

**Figure 11.** Pressure field in the driven gas for the heavy inhomogeneity ($A = 0.51$) case at different dimensionless times.

For clarity, Figures 10 and 11 show the pressure field only in the driven gas, while Figure 12 shows the pressure field both outside and inside the inhomogeneity. Our calculations also capture the twin regular reflection-refraction (TRR) configuration (Figure 12), firstly observed by Henderson [26]. The TRR is a four-shock configuration where the refracted shock moves faster than the incident shock, with the reflected shock moving in the opposite direction. The fourth shock is the side shock, which connects the refracted shock with the incident shock (Figure 12b). The pressure field in some of the figures is slightly "noisy" at certain locations. This is probably the result of the piecewise constant smoothing length used in this study. However, this occurs far from the gas inhomogeneity and does not affect the inhomogeneity shape evolution analysis carried out in this study.

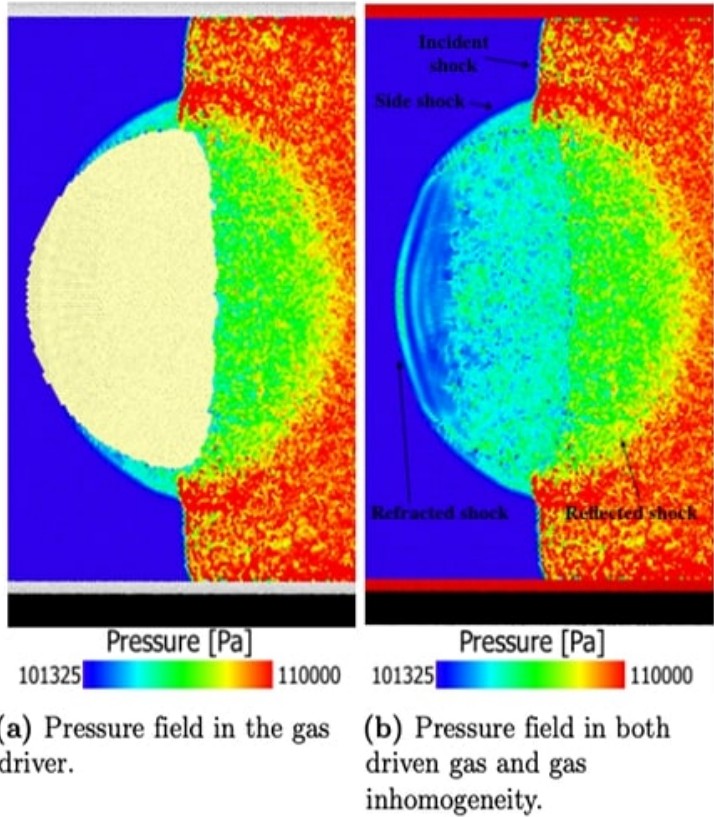

**(a)** Pressure field in the gas driver.

**(b)** Pressure field in both driven gas and gas inhomogeneity.

**Figure 12.** Twin regular reflection-refraction (TRR) in the light inhomogeneity case ($A = -0.79$) at $\tau = 0.88$.

Refracted and Reflected Waves (Acoustic Lens and Acoustic Mirror)

When the shock passes through the light inhomogeneity, the speed of the wave increases, while its pressure decreases. As a result of this, the direction of the refracted wave diverges following a direction given by the high-pressure region behind the light inhomogeneity (Figure 10b). Conversely, when the pressure wave passes through the heavy inhomogeneity, the speed decreases and the pressure increases; in this case the refracted wave converges to the high-pressure region in front of the heavy inhomogeneity (Figure 11b). This behaviour suggests that a gas inhomogeneity could behave, to a certain degree, like an acoustic lens.

When a collimate beam of light passes through an optical lens, the direction of the beam changes according to the position of the focal point of the lens. In divergent lenses (Figure 13a), the focal point is behind the lens; in convergent lenses (Figure 13b), it is in front of the lens, similar to a collimate beam of

light diverged or converged. With respect to the focal point, the shock wave is either diverged or converged with respect to the high pressure region area discussed in the previous section (compare Figures 10b and 11b with Figure 13c,d). Considering, therefore, that the focal point and the high pressure region play a similar role, it is possible to identify the high pressure region as a focal region of the system.

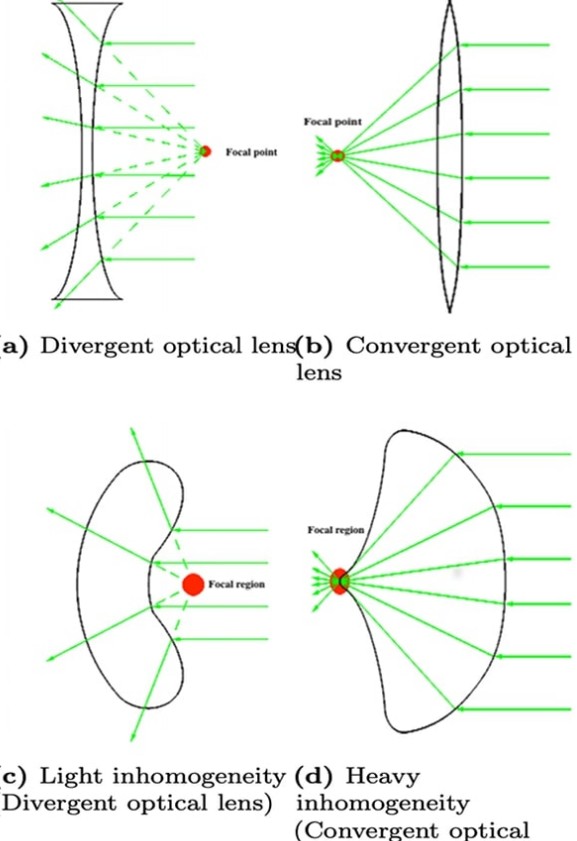

**(a)** Divergent optical lens **(b)** Convergent optical lens

**(c)** Light inhomogeneity (Divergent optical lens) **(d)** Heavy inhomogeneity (Convergent optical lens)

**Figure 13.** Comparison between optical lenses and gas inhomogeneities behaving as acoustic lenses.

Another interesting observation concerns the shape of the reflected wave (Figures 10 and 11). Observing the pressure field, it is possible to see how the evolution of the reflected wave mirrors reflects, to a certain degree, that of the inhomogeneity. In the light inhomogeneity, the mirror image (Figure 10c) has a shape similar to the semi-claw standard shape, it is just more elongated in the $y$-direction. In the heavy inhomogeneity, the mirror image (Figure 11d) has a shape similar to the jellyfish head standard shape.

## 7. Conclusions

Typically, in gas dynamics with SPH, the approach is to employ a smoothing length varying with density. In this paper we show that a simpler piecewise constant smoothing length is sufficient to model the dynamics of inhomogeneities in shock tubes. With this device, which fits the underlying physics, we obtain (i) the correct shapes, (ii) the correct timescale and (iii) the correct refraction/reflection of the wave (something CFD simulations sometimes fail to achieve). This can be useful in at least two directions. Firstly, in combination with the energy conservation equation, the proposed SPH approach could be adapted for simulating a variety of phenomena related with the so-called Richtmyer–Meshkov instability [9] such as supersonic mixing and gas combustion in Scramjet. Secondly, it can be integrated with

Discrete Multiphysics (DMP) for the simulation of cavitation erosion. Discrete Multi-Physics (e.g., [27,28]) is a multiphysics technique that, contrary to traditional multiphysics, is based on computational particles rather than computational meshes. It combines different particle-based modelling techniques such as smooth particle hydrodynamics, discrete element method and the lattice spring model, and it has been effectively used for fluid-structure interaction problems (e.g., [29–31]). DMP, in particular, is superior to traditional multiphysics in the case of phase-transition [32], agglomeration [33] and break-up of solid structures [34]. Specifically, the SPH model presented in this study, in particular, could be coupled with the break-up module in DMP to model cavitation generated shock waves and their effects, including erosion, on nearby solid surfaces.

**Supplementary Materials:** The following are available online at http://www.mdpi.com/2076-3417/9/24/5435/s1, Video1: Heavy inhomogeneity evolution with pressure field, Video2: Light inhomogeneity evolution with pressure field.

**Author Contributions:** A.A. (Andrea Albano) and A.A. (Alessio Alexiadis) conceptualise the work; A.A. (Andrea Albano) designed the work and performed the simulations; A.A. (Andrea Albano) and A.A. (Alessio Alexiadis) contributed in writing–review and editing the paper.

**Funding:** This work was supported by the US Office of Naval Research Global (ONRG) under 256 NICOP Grant N62909-17-1-2051.

**Conflicts of Interest:** The authors declare no conflict of interest

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
