# Peer review of "Interaction of Shock Waves with Discrete Gas Inhomogeneities: A Smoothed Particle Hydrodynamics Approach"

_applsci, doi:10.3390/app9245435_

Round 1

Reviewer 1 Report

The paper concerns numerical simulation of shock wave interaction with a cylindrical gas inhomogeneity. Smoothed Particle Hydrodynamics (SPH) technique was chosen for calculations. The comparison between experimental and simulation results demonstrates the applicability of the SPH method for adequate description of the flow field around inhomogeneity. The simulation results also reproduce the dynamics of deformation of the inhomogeneity upon interaction with shock wave. Particular disadvantage of this work is relatively short overview of the previous works.

Some remarks should be addressed before final submission:

For overview completeness it is necessary to include the following recent references: [1] S.P. Medvedev et al. Interaction of blast waves with helium-filled rubber balloons. 2019 J. Phys.: Conf. Ser.1147 012021. [2] P.Yu. Georgievskiy, V.A. Levin,·O.G. Sutyrin Interaction of a shock with elliptical gas bubbles. Shock Waves (2015) 25:357–369 DOI 10.1007/s00193-015-0557-4 The simulations were performed under 2D approach, while in many cases an inhomogeneity has a 3D shape (for example sphere). What is difference between 2D and 3D simulation results? The authors (even in Title) introduce a lot of “new” terms like “shock-wave”, “shock-tube”, “Smoothed-Particle”, etc. The reviewer strongly recommends excluding hyphens that are not relevant to common standards. The detailed description of the shock tube principals in Introduction section is not necessary for this paper, since it is well known to specialists.

Author Response

Reviewer 1

For overview completeness it is necessary to include the following recent references: [1] S.P. Medvedev et al. Interaction of blast waves with helium-filled rubber balloons. 2019 J. Phys.: Conf. Ser.1147 012021. [2] P.Yu. Georgievskiy, V.A. Levin,·O.G. Sutyrin Interaction of a shock with elliptical gas bubbles. Shock Waves (2015) 25:357–369 DOI 10.1007/s00193-015-0557-4

Answer:

Those references have been added to the revised version of the manuscript (line 12)

The simulations were performed under 2D approach, while in many cases an inhomogeneity has a 3D shape (for example sphere). What is difference between 2D and 3D simulation results?

Answer:

As written in line 25 of the submitted manuscript, a gas inhomogeneity can be experimentally created with a cylindrical shape. In the work of Haas and Sturtevant (1987) they used a cylindrical gas inhomogeneity that can be successfully modelled by a 2D model as shown by Quirk and Karni (1997) and our study.

A new figure, Figure 2, is added to clarify the geometry of the gas inhomogeneity and the shock tube.

The authors (even in Title) introduce a lot of “new” terms like “shock-wave”, “shock-tube”, “Smoothed-Particle”, etc. The reviewer strongly recommends excluding hyphens that are not relevant to common standards.

Answer:

Corrected.

The detailed description of the shock tube principals in Introduction section is not necessary for this paper, since it is well known to specialists.

Answer:

The shock tube description has been simplified. However, we believe the explanation of the shock tube helps understanding the geometry of the cylindrical gas inhomogeneity and, for this reason kept in the text

Reviewer 2 Report

The authors have employed SPH modelling to simulate a shock-wave interacting with a cylindrical

air inhomogeneity in a shock tube. The manuscript is well-written and the model and

presentation of results suitable for publication. Results are sound and itneresting and

references to the literature appropriate. 

While I am happy to recommend its publication, I think that a few minor things can

add to the quality of the manuscript before publication.

Minor comments:

1) Title should be ‘…: an SPH… ‘ instead of ‘…: a SPH’. It is probably better to have the full term ‘Smoothed Particle Hydrodynamics’ in the title.

2) Abstract —>  please correct:  ‘… interacting with a cylindrical gas inhomogeneities…’

—> ‘The gas inhomogeneity, …. ‘: Improve this sentence

—> It would be good to mention on the results that the method is validated against CFD reslults. ‘Accurate’, in terms of what?

Discussion should be expanded in abstract to include a few highlights in the manuscript and also include new elements of their modelling approach.

3) Intro: 

—> ‘a benchmarks…’ : please correct

4) Section 2: various sentences require editing to be grammatically correct

5) an overall update of text in the whole manuscript will be greatly beneficial.

6) I would ask the authors to add a bit more detail on SPH in the methods section, 

including a bit more technical detail rather than general aspects of the theory. 

Author Response

Reviewer 2

1) Title should be ‘…: an SPH… ‘ instead of ‘…: a SPH’. It is probably better to have the full term ‘Smoothed Particle Hydrodynamics’ in the title.

Answer:

We changed the title and used ‘…: a Smoothed Particle Hydrodynamics’

2) Abstract —>  please correct:  ‘… interacting with a cylindrical gas inhomogeneities…’

—> ‘The gas inhomogeneity, …. ‘: Improve this sentence

—> It would be good to mention on the results that the method is validated against CFD reslults. ‘Accurate’, in terms of what?

Discussion should be expanded in abstract to include a few highlights in the manuscript and also include new elements of their modelling approach.

Answer:

Abstract and text modified according to the reviewer suggestions.

3) Intro: 

—> ‘a benchmarks…’ : please correct

Answer:

The typo has been corrected.

4) Section 2: various sentences require editing to be grammatically correct

5) an overall update of text in the whole manuscript will be greatly beneficial.

Answer:

Done, as suggested.

6) I would ask the authors to add a bit more detail on SPH in the methods section, 

including a bit more technical detail rather than general aspects of the theory. 

Answer:

Section 5 has been expanded and improved, as suggested